# Characterization and Biocompatibility Properties In Vitro of Gel Beads Based on the Pectin and κ-Carrageenan

**DOI:** 10.3390/md20020094

**Published:** 2022-01-22

**Authors:** Sergey Popov, Nikita Paderin, Daria Khramova, Elizaveta Kvashninova, Anatoliy Melekhin, Fedor Vityazev

**Affiliations:** Institute of Physiology of Federal Research Centre “Komi Science Centre of the Urals Branch of the Russian Academy of Sciences”, 50 Pervomaiskaya Str., 167982 Syktyvkar, Russia; paderin_nm@mail.ru (N.P.); dkhramova@gmail.com (D.K.); kvashninova.e@yandex.ru (E.K.); melekhin-anatoliy@yandex.ru (A.M.); rodefex@mail.ru (F.V.)

**Keywords:** apple pectin, κ-carrageenan, gel beads, protein adsorption, complement activation, haemolysis, peritoneal macrophages, TNF-α, NF-κB, TLR4, ICAM-1

## Abstract

This study aimed to investigate the influence of kappa (κ)-carrageenan on the initial stages of the foreign body response against pectin gel. Pectin-carrageenan (P-Car) gel beads were prepared from the apple pectin and κ-carrageenan using gelling with calcium ions. The inclusion of 0.5% κ-carrageenan (Car0.5) in the 1.5 (P1.5) and 2% pectin (P2) gel formulations decreased the gel strength by 2.5 times. Car0.5 was found to increase the swelling of P2 gel beads in the cell culture medium. P2 gel beads adsorbed 30–42 mg/g of bovine serum albumin (BSA) depending on pH. P2-Car0.2, P2-Car0.5, and P1.5-Car0.5 beads reduced BSA adsorption by 3.1, 5.2, and 4.0 times compared to P2 beads, respectively, at pH 7. The P1.5-Car0.5 beads activated complement and induced the haemolysis less than gel beads of pure pectin. Moreover, P1.5-Car0.5 gel beads allowed less adhesion of mouse peritoneal macrophages, TNF-α production, and NF-κB activation than the pure pectin gel beads. There were no differences in TLR4 and ICAM-1 levels in macrophages treated with P and P-Car gel beads. P2-Car0.5 hydrogel demonstrated lower adhesion to serous membrane than P2 hydrogel. Thus, the data obtained indicate that the inclusion of κ-carrageenan in the apple pectin gel improves its biocompatibility.

## 1. Introduction

The construction of scaffold materials for wound healing and tissue engineering based on hydrogels is a recent approach based on the similarity of the structure and functions of hydrogels to the extracellular matrix [1,2]. Natural polymers for hydrogel preparation have advantages such as bioactivity, low toxicity, and biocompatibility compared to synthetic ones [3,4]. Plant polysaccharides are potential candidates for tissue engineering scaffolds due to their gelling properties and biocompatibility [5,6]. Pectin is a heteropolysaccharide that is part of the cell walls of higher plants, which is widely used in the pharmaceutical and food industries [7,8]. Pectins include a diverse group of polysaccharides whose backbone is formed by 1,4-linked α-D-galacturonic acid (GalA) residues that can be partly substituted with methyl ester at C6-carboxyl groups. Pectic macromolecule generally consists of homogalacturonan (HG), rhamnogalacturonan-I (RG I), rhamnogalacturonan-II, and xylogalacturonan. RG I has a backbone of repeating disaccharides of GalA and rhamnose (Rha) with the side chains of arabinose (Ara) and galactose (Gal) residues. Rhamnogalacturonan-II consists of a backbone of HG with complicated side chains containing simple sugars linked to the GalA [9]. Pectin with a low degree of methyl esterification forms a gel in the presence of divalent cations such as calcium ions. Pectin gelation includes an initial dimerization of two slightly shifted chains and negligible subsequent aggregation of these dimers. Pectin egg-box dimers are assumed to grow in a “dotting” mode due to a block-wise and a random distribution of non-methoxylated GalA units in pectin molecules [10,11,12].

Biomaterials based on pectin have been proposed for tissue engineering [13,14], wound dressing [15,16], drug delivery [17], and other biomedical applications [8]. Pectin shows good cytocompatibility with fibroblasts [18,19,20] and other cells [21,22,23], suggesting that pectin scaffolds are biocompatible. However, it is worth mentioning that pectins can bind blood proteins [24], including proteins of the complement system [25,26,27], and enhance the production of pro-inflammatory cytokines and nitric oxide by macrophages [28,29,30,31]. It is well known that non-specific protein adsorption, complement activation, and pro-inflammatory activity of macrophages, are signs of the initial stage of a foreign body response (FBR) that develops against the implanted material [32]. Subsequent transformation of macrophages into foreign body giant cells and fibrosis or fibrous capsule formation by activated fibroblasts results in poor integration implant with native tissue [32]. Therefore, pectin hydrogel may present challenges in tissue engineering due to a FBR. The targeted design of new biomaterials based on pectin appears to require characteristics inhibiting protein adhesion, complement activation, and subsequent activation of inflammatory macrophages.

Carrageenans are formed of alternate units of D-Gal and 3,6-anhydro-Gal joined by α-1,3- and β-1,4-glycosidic linkage. Kappa-(κ), iota-, and lambda-carrageenan differ in the sulphate content and are of current commercial importance [33]. Carrageenans have been shown to demonstrate a broad spectrum of biological activity, which depends on the structure, such as antiviral, anticoagulant, antitumor, immunomodulatory, and other activity [34]. The κ-type carrageenan forms a brittle gel with potassium ions and gives a soft, elastic gel on interaction with calcium ions [35]. Recently, many studies have been devoted to κ-carrageenan-based hydrogels for biomedical applications, including drug delivery and wound dressing [35,36,37]. Generally, κ-carrageenan is assumed to resemble native sulphated glycosaminoglycans and demonstrates good biocompatibility. Therefore, in the present study, κ-carrageenan was selected to increase the biocompatibility of pectin hydrogel. Several types of pectin-carrageenan materials have been previously studied [38,39,40]. However, the effect of carrageenan on the ability of pectin gel to induce FBR has not been investigated.

The aim of the study is to develop pectin-carrageenan gels with improved biocompatibility. For this, hydrogels were obtained by blending apple pectin and κ-carrageenan using cross-linking with calcium ions. Protein adsorption, complement and macrophages activation, haemolysis rate, and adhesion to serous tissue were analysed to determine the effect of κ-carrageenan inclusion on the ability of pectin hydrogel to initiate FBR.

## 2. Results and Discussion

### 2.1. Rheological Properties of Pectin-Carrageenan Hydrogel

The viscoelastic properties were studied using oscillatory assessments. We recorded the parameters of storage or elastic modulus (G’) that shows the solid-like properties of the material, and the loss or viscous parameters modulus (G″) that shows the liquid-like or viscous characteristics of the material [41,42] The mechanical spectra of P-Car hydrogels, which are based on the variation of G’ and G’’ with frequency, are illustrated in Figure 1A–C and Table 1.

The G’ values of pectin hydrogels increased with increasing pectin concentration from 1 to 4% (Figure 1A–C). Adding 0.5% κ-carrageenan to 1% pectin gel significantly increased G’. The G’ range was 520 to 1190 Pa and 1450 to 2260 Pa for the P1 and P1-Car0.5 hydrogel, respectively (Figure 1A). In the case of 1.5% pectin gel, the addition of κ-carrageenan reduced G’. The G’ range was 994 to 1910 Pa and 678 to 1180 Pa for the P1.5 and P1.5-Car0.5 hydrogel, respectively (Figure 1B). Unlike 1 and 1.5% pectin gel, adding κ-carrageenan to 2% pectin gel did not change the G’ value. The range of G’ values was from 1163 to 1790 Pa and from 1330 to 1790 Pa for the P2 and P2-Car0.5 hydrogel, respectively (Figure 1C).

The loss factor tan (δ), which was calculated from the ratio of G″ to G′ and varied from zero to infinity, represents the tendency of the material to liquid or solid-like behaviour. A confirmation of the viscoelastic state of gels cross-linked by ion Ca^2+^ was obtained when tan δ = 1 (G’ = G’’) (Figure 1D–F). All the hydrogels prepared had the tan δ values at low frequencies close to 1, further increasing the frequency tan δ ≥ 1.

The data obtained indicate that P and P-Car hydrogels possess a “viscoelastic” structure [43,44,45]. The effect of adding κ-carrageenan on the rheological properties depends on the concentration of pectin in the blended gel. In subsequent experiments, only P1.5 and P2 based gels were used because P1-based gels were weak.

### 2.2. Preparation and Characterization of Gel Beads

Pectin-carrageenan (P-Car) hydrogel beads were prepared by a simple solution mixing of apple pectin (P) and κ-carrageenan (Car) followed by dropwise adding the mixture to calcium chloride solution (Figure 2A).

The weight, projected equivalent diameter, and gel strength of gel beads are shown in Table 2. P2 gel beads had a weight 30% higher and a diameter 33% lower than P1.5 beads. Accordingly, P2 gel beads were found to be of higher density and strength compared to P1.5 beads. The inclusion of κ-carrageenan into both 2 and 1.5% pectin gel failed to affect bead size. However, κ-carrageenan significantly increased the weight and reduced the strength of the gel beads.

The strength of P2-Car0.2 and P2-Car0.5 gel beads was found to be 1.9 and 2.6 times lower, respectively, than that of P2 beads. The strength of P1.5-Car0.5 gel beads was 2.5 times lower than that of P1.5 beads. Gelling mechanism of low methyl esterified pectin is well known to involve an interaction of a high number of free carboxylic acid groups (COO^−^) with divalent ions forming strong links between the pectin chains (Figure 2B) [12]. Therefore, an increase in the concentration of pectin increased the gel density and strength. The decrease in gel strength by κ-carrageenan is consistent with previously obtained data [39]. Adjacent κ-carrageenan chains form double helices with outward-oriented sulphate (OSO_3_^−^) groups, the cross-linking of which leads to the formation of a three-dimensional network [46]. The binding of Ca^2+^ by these sulphate groups of κ-carrageenan may competitively inhibit the cross-linking of pectin chains (Figure 2B). In addition, decreasing strength of the pectin-carrageenan gel may also be originated from the high water absorption ability of κ-carrageenan due to polar OSO_3_^−^ groups [47].

### 2.3. Swelling Studies

Swelling is a significant factor in the adsorption properties and stability of hydrogels. In this study, the swelling degree of the gel beads incubated in phosphate-buffered saline (PBS), distilled water, and cell culture medium was investigated (Figure 3). 

The swelling degree of P2 gel beads after 2 h of incubation in PBS was 300% (Figure 3A). However, the destruction of the gel beads was detected after 4 h of incubation in PBS. P2 gel beads swelled in distilled water to 100% in 4 h and then retained their size during 24 h of incubation (Figure 3B). The swelling degree of P2 gel beads was 180, 300, and 350% after 4, 8, and 24 h of incubation in cell culture medium (Figure 3C). In PBS, the possible reason for bead disintegration can be the displacement of calcium ions that crosslink the hydrogel matrix with sodium, potassium, and phosphate ions of PBS, resulting in water uptake, rapid swelling, and hydrogel erosion [48]. The stability of pectin gel beads in water was higher than in PBS, due to the lack of intense ion exchange. 

Hank’s balanced salt solution containing 1.3 mM calcium ions was used as a cell culture medium. Therefore, the mechanisms of the external gelling of pectins can be considered to interpret the properties of gel beads in the cell culture medium. Two-stage external gelling of pectin is accompanied by the simultaneous aging of the gel and the formation of an inhomogeneous structure of the gel, even after a very long time [49]. The initial phase of pectin gelation at calcium solution perfusion may last 2–3 h until the gel behaves like an arrested gel structure [49]. In our study, at preparation, gel beads were allowed to stand in a calcium chloride solution for only 30 min before separation and washing. Therefore, the calcium ions may be supposed to diffuse from the cell culture medium and increase cross-linking and the strength of the gel. However, to estimate the possible rate of calcium ion diffusion from the medium requires the determination of the calcium concentration in the gel body, which was not carried out. In the study of Secchi et al. [49], it was found that pectin gels are much less stressed in the presence 70 mM NaCl, probably due to the reduction the rigidity of the polymer chains. According to these data, NaCl contained in the cell culture medium at a concentration of 150 mM can also hinder the local internal stresses and reduce gel restructuring. The diameter of the gel beads studied was 1–1.5 mm, while a dense and homogeneous gel is formed at a depth of 7 mm from the source of calcium ions [49]. Therefore, the effect of gel heterogeneity in the present study may be insignificant.

The inclusion of κ-carrageenan into pectin gel failed to affect the swelling behaviour of gel beads in PBS and distilled water (Figure 3A,B). However, κ-carrageenan increased the swelling of pectin gel beads significantly in the cell culture medium. The swelling degree of P2-Car0.5 beads was higher by 2.0, 1.4, and 1.4 times than P2 beads, respectively, after 4, 8, and 24 h of incubation in cell culture medium (Figure 3C). These data are consistent with a high degree of water absorption by the carrageenan-containing gel in simulated body fluids [39,50]. The hydrophilicity of carrageenan appears to provide a high swelling capacity of the pectin-carrageenan gel beads. The hydrophilicity of pectin-κ-carrageenan films increased with increasing κ-carrageenan content [51]. The swelling ratio of alginate-carrageenan films increased with gradual increasing carrageenan content [47]. 

### 2.4. Protein Adsorption

The adsorption of proteins is the first event in the processes occurring when blood contacts a ‘‘foreign” surface in a tissue engineering scaffold, medical device, or biomaterial [52]. Although the proteins present on most investigated surfaces include multi-protein systems including albumin, fibrinogen, immunoglobulins, vitronectin, etc., an understanding of the adsorption of a single protein is essential to understand the adsorption behaviour of individual gel materials. Serum albumin represents the most abundant proteins in human plasma (human serum albumin 40 mg/mL [53]), excluding haemoglobin. BSA was selected as a model protein to evaluate protein adsorption due to its chemical similarity to human serum albumin. The BSA adsorption of P-Car gel beads is shown in Figure 4.

P2 gel beads were found to adsorb 30–42 mg/g of BSA depending on pH. The adsorption capacity of gel beads was found to be higher at pH 3 than at pH 5 and 7. Interaction forces that participate in the adsorption of BSA on the gel can be attributed to the ionic electro-static attraction, van der Waals interactions, hydrogen bonding, and hydrophobic interactions. The adsorption of BSA on hydrogel through electrostatic interactions can be explained by the presence of pH-sensitive carboxyl groups of pectin on the surface of gel beads. The presence of negatively charged carboxyl groups on the surface of the pectin gel may promote protein adsorption through electrostatic interactions when the pH of the solution is lower than the isoelectric point of BSA (pI ~4.7) and protein is positively charged. Accordingly, electrostatic repulsion may explain a decrease of BSA adsorption at pH 7 due to BSA having a tendency to present a high net negative density charge, producing a high repulsion on the gel surface at pH 7.

The inclusion of κ-carrageenan into apple pectin gel led to a decrease in protein adsorption at all pH values tested. However, the greatest decline in BSA adsorption by P-Car gel beads compared to pure P gel beads was observed at pH 7. P2-Car0.2, P2-Car0.5, and P1.5-Car0.5 beads were found to reduce BSA adsorption by 3.1, 5.2, and 4.0 times compared to P2 beads, respectively, at pH 7. Sulphate groups of κ-carrageenan seem to lead to a higher negative charge in the final gel matrix [54]. The inclusion of κ-carrageenan into apple pectin gel decreased protein adsorption to a lesser extent in an acidic than in the neutral medium. P2-Car0.2, P2-Car0.5, and P1.5-Car0.5 beads were found to reduce BSA adsorption by 2.4, 3.5, and 2.8 times compared to P2 beads, respectively, at pH 5. P2-Car0.2, P2-Car0.5, and P1.5-Car0.5 beads were found to reduce BSA adsorption by 2.3, 3.5, and 2.5 times compared to P2 beads, respectively, at pH 3. 

The results are consistent with previous data that κ-carrageenan prevents nonspecific adhesion of blood proteins. Films prepared of 2% κ-carrageenan using KCl have been shown to adsorb a low amount of BSA at incubation for 2 h at pH 5 and 7.4 [54,55]. In the study [56], 2% κ-carrageenan gel beads were found to adsorb about 0.9 mg/g BSA at incubation for 1 h in PBS. The lower values compared to our results (12–20 mg/g for P-Car) can be explained by a shorter incubation time and an additional procedure for washing the beads with 2% sodium dodecyl sulphate in the mentioned study. Lokhande et al. 2018 [57] demonstrated low protein adsorption on carrageenan beads using a fluorescein isothiocyanate-labelled bovine serum albumin solution. 

### 2.5. Complement Activation

The complement system is comprised of more than 30 proteins that interact with immune cells in blood to help them identify and clear ‘foreign’ bodies. It is well-known that biomaterials in contact with blood can activate complement leading to recruitment, activation, and binding of leukocytes to the surface [52]. Direct binding of complement proteins, the nonspecific adsorption of antibodies (leading to the classical activation mechanism), and the accumulation of C3b (leading to the alternative activation mechanism) have all been proposed as possible initiators of complement activation by biomaterials surfaces [58]. Complement activation is a clinically relevant biological property representing one of the parameters included in the criteria for testing the hemocompatibility of biomaterials defined in ISO 10993-4. 

Complement component C3 is known to undergo a surface-induced change in conformation, which can induce alternative activation of the complement system leading to the release of reactive fragments C3a, C5a and SC5b-9 to plasma [59]. Here, the release of C3a was measured in human blood after incubation of the gel beads in blood samples in vitro to analyse the capacity of the hydrogels to activate the alternative complement cascade (Figure 5A).

The C3a level in the blood samples incubated with P2 gel beads was 2.7-fold higher than the C3a level of blood sample treated with saline taken as a negative control (Figure 5B). However, the level of C3a produced by P2 beads was lower, 2-fold, compared to blood samples treated with zymosan (positive control). The results indicate that apple pectin hydrogel promoted a slight, but significant, release of C3a as compared to the initial blood sample. 

These results are in agreement with the data that pectins express complement fixation activity through both alternative and classical pathways [25,26]. In particular, Wang et al. (2016) [60], Michaelsen [61], Kiyohara [62], and others found that pectins interact directly with the C3 complement proteins. The polysaccharides consisting of more linear HG regions have been reported to possess lower complement fixation activity compared to the polysaccharides consisting of more ramified regions. Accordingly, the higher amounts of Ara- and Gal- residues in the side chains of branched regions may result in high complement fixating activity of pectin [63,64,65]. The content of GalA residues in the apple pectin used in the present study for the gel beads is 87%, indicating a significant proportion of linear HG. However, a small amount of Ara and Gal residues of the RG-I region probably mediated interaction pectin gel beads with complement proteins. Earlier, we found a slight activation of the C3a protein caused by 1 and 4% apple pectin gel prepared by mixing a heated pectin solution and a frozen calcium chloride solution [27]. 

It is important to note that 0.5% κ-carrageenan solution, unlike 2% apple pectin solution, failed to activate C3a production (Figure 5B). The levels of C3a activation on P2-Car0.2 and P2-Car0.5 beads were comparable with that on P2 gel beads, whereas P1.5-Car0.5 beads activated complement to a lesser extent. The latter seems to have a higher proportion of sulphate groups from κ-carrageenan concerning the amount of Ara and Gal residues of pectin moiety. Carrageenans, being sulphated polysaccharides, appear to mimic “host polyanions” such as sulphated glycosaminoglycans in inhibition of complement. Carrageenans have been recently shown to inhibit C3 binding to plate wells coated with LPS depending on the sulphation degree [66]. Composite hydrogels obtained from carrageenan crosslinked with poly-(acrylic acid) [56] and gel composed from carrageenan and chitosan [67], have been earlier found to decrease the formation of C3a. It is generally accepted that biomaterial-induced complement activation is initiated by free hydroxyl (-OH) groups on the material’s surface, as they promote the formation and covalent binding of the cleaved form of C3, C3b [68]. According to our data, additional sulphate groups of the pectin-carrageenan gel appears to reduce the proportion of -OH groups resulting decrease in binding and activation of the complement system.

### 2.6. Haemolysis Assay

The haemocompatibility was evaluated using the measurement of haemoglobin release after incubation of P-Car gel beads with whole blood. The levels of haemolysis while human blood interacts with P and P-Car gel beads are given in Table 3. Haemolysis by P2 and P2-Car0.2 gel beads was equal to 2.6 and 2.1%, respectively. Pectin gel prepared by mixing a heated pectin solution and a frozen calcium chloride solution induced a 4.8 ± 0.7 haemolysis rate [27]. Many other studies have previously shown good haemocompatibility of pectin-based biomaterials [69,70,71].

P2-Car0.5 and P1.5-Car0.5 beads were found to cause haemolysis 1.5 times less than P2 beads (Table 3), indicating that the inclusion of κ-carrageenan into pectin gel decreased erythrocytes haemolysis. These results are in agreement with the data that both κ-carrageenan and κ-carrageenan-based materials induced negligible haemolysis and failed to change the morphology of red blood cells [56,72,73]. The results obtained indicate that although the pure pectin gel demonstrates good haemocompatibility, κ-carrageenan further reduces the risk of possible damage to erythrocyte membranes.

### 2.7. Serosal Adhesion of Hydrogels

Bioadhesion is a significant characteristic of implantable materials as it affects the inside localization of the implant and its affinity for the surrounding tissues. Although the mucoadhesive properties of pectin are widely used in oral delivery systems, only a few studies have focused on the adhesion of pectin gels to the serosa [24,74,75]. The measurement results for hydrogels adhesion to the serous membrane are shown in Figure 6.

In this experiment, hydrogels formed at the tip of a cotton swab were brought into contact with a sample of the inner surface of rat abdominal wall fixed to a platform, and the force required to detach them was recorded (Figure 6A–C). The hydrogel “head” and a corresponding thick patch of serosa were compressed at 0.05 N for 20 s before probe withdrawal. Separation of the surfaces was not associated with serosa rupture and residual tissue could not be observed adhering to the hydrogel surface after the withdrawal. The P2-Car0.5 hydrogel demonstrated lower adhesion force and failed to influence the work of adhesion. P1.5-Car0.5 hydrogel did not change the adhesion force and work of adhesion (Figure 6D).

The mechanism of pectin adhesion to the serosa may resemble the entanglement or interpenetration of pectin chains and cellulose microfibrils in a plant cell wall [9]. It has been shown that restrictive cross-linking of pectin in the plant wall reduces its adhesiveness. Therefore, cross-linking of the pectin and κ-carrageenan chains in blended hydrogels can diminish the possibility of physical entanglement of the chains between the gel and serosa. Moreover, an additional negative charge due to sulphate groups of carrageenan is supposed to increase the strength of the electrostatic repulsion between the hydrogel surface and the negatively charged layer of the serous membrane. It is interesting to note that films of high methyl esterified citrus pectin have been found to demonstrate significantly greater adhesivity to the visceral pleura surface than that of low methyl esterified pectin [75,76,77]. Low methyl esterified pectin contains more negatively charged carboxyl (COO^-^) groups than high methyl esterified pectin. Therefore, electrostatic repulsion plays a significant role in the adhesion processes between the blended hydrogel surface and the serous membrane.

### 2.8. Peritoneal Macrophages Adhesion and Activation

Following the initial surface protein matrix formation and complement activation, acute inflammation is provoked, where polymorphonuclear leukocytes and macrophages are responsible for the destruction and integration of biomaterials [78]. In the present study, the gel beads were co-incubated with a suspension of mouse peritoneal macrophages. The number of macrophages adsorbed by the beads was calculated as the difference between the initial number of cells and the number of cells remaining free in the culture medium after co-incubation for 4 h. The levels of the proinflammatory tumour necrosis factor-α (TNF-α and anti-inflammatory interleukin-10 (IL-10) in the supernatant indicated the degree of activation of the peritoneal macrophages. The number of peritoneal macrophages adhered on the P1.5-Car0.5 gel beads was about 28% fewer than on the P2 and P2-Car0.5 beads (Figure 7A). Moreover, macrophages produced significantly less TNF-α in the presence of P1.5-Car0.5 than P2 and P2-Car0.5 gel beads (Figure 7B). No differences in IL-10 levels were found (Figure 7C). A decrease in TNF-α production by macrophages was accompanied by a decrease 1.7 fold in the expression of nuclear factor kappa B (NF-κB), which is an intracellular messenger of the inflammatory cascade (Figure 8A,B).

Activation of macrophages can occur both when interacting with a provisional protein matrix or as a result of direct contact of cells with the gel surface [32]. In the first case, the proteins of foetal bovine serum, which are part of the cell culture medium, seem to be the first to adhere to the gel beads and act as a trigger of cell activation. Therefore, the reduced protein adsorption capacity of P-Car gels provides less adhesion and proinflammatory activation of peritoneal macrophages. In the second case, toll-like receptors (TLRs), integrins and scavenger receptors have been suggested to mediate recognizing of gel surfaces by polymorphonuclear leukocytes and macrophages in the early phase of inflammation [32]. The TLR4 and inter-cellular adhesion molecule 1 (ICAM-1) protein levels in peritoneal macrophages incubated with gel beads were measured using Western blotting. As shown in Figure 7, P1.5-Car0.5 gel beads decreased TLR4 level by 238% (Figure 8C). However, the differences between the groups were not significant. No differences in ICAM-1 levels were found (Figure 8D). Thus, the data obtained indicate that the reducing effect of k κ-carrageenan on the inflammatory response of macrophages to gel beads may be mediated by a decrease in protein adsorption. The results are consistent with previous data by Pettinelli et al. [79] that κ-carrageenan hydrogels failed to induce NO production by macrophages and did not allow cell attachment. Kappa-carrageenan wrapped nanoparticles have shown good anti-inflammatory activity in vitro [80].
Figure 7The ability of pectin-carrageenan gel beads to adhere to the peritoneal macrophages (**A**); to stimulate production of TNF-α (**B**) and IL-10 (**C**). Results are presented as the mean ±S.D. (*n* = 8). *—*p* < 0.05 vs. P2.
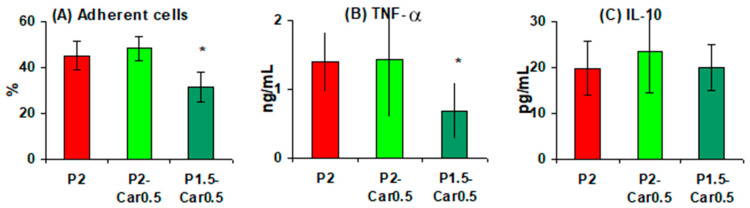

Figure 8The impact of gel beads on NF-κB, TLR4, and ICAM-1 in peritoneal macrophages. Representative bands of NF-κB, TLR4, ICAM-1, and β-actin (**A**). Bar graph showing NF-κB (**B**), TLR4 (**C**), and ICAM-1 (**D**) protein levels, which are normalized to the β-actin protein level. The results are presented as the mean ±S.D. (*n* = 8) * *p* < 0.05 vs. P2.
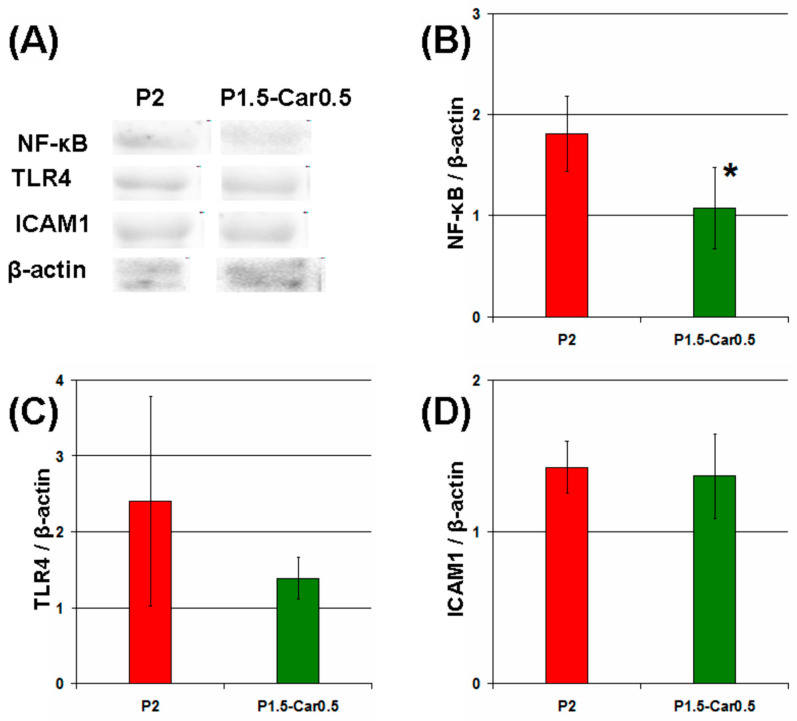


## 3. Materials and Methods

### 3.1. Polysaccharides

Apple pectins AU701 was purchased from Herbstreith and Fox (Nuremberg, Germany). Refined KA120R κ-carrageenan was purchased from Greenfresh food Co., Ltd (Fujian, China). The chemical characteristics of the pectin and κ-carrageenan are shown in Table 4.

The content of monosaccharides was detected by gas-liquid chromatography (GLC) in a Varian 450-GC (Uden, The Netherlands) chromatograph after hydrolysis of the polysaccharides and transformation of monosaccharides into the corresponding alditol acetates. The homogeneity and molecular weight of the polysaccharide samples were determined by high performance gel-permeation chromatography (HPGPC) on a chromatographic system for the analysis included an LC-20AD pump (Shimadzu, Tokyo, Japan), a DGU-20A3 degasser (Shimadzu, Tokyo, Japan), a CTO-10AS thermostat (Shimadzu, Tokyo, Japan), a RID-10A refractometer (Shimadzu, Tokyo, Japan) as a detector, a PPS SUPREMA 3000A 10 µm (8.0 MM × 300 MM) and a PPS SUPREMA 10 µm (8.0 MM × 50 MM) (PSS, Amherst, MA, USA). Pullulans (1.3, 6, 12, 22, 50, 110, 200, 400, and 800 kDa) were used as standards. The detailed procedure was described earlier [81]. Dodgson’s method with the calibration curve for potassium sulphate was used for the content of sulphate groups [82]. Spectrophotometric measurements were made with an Ultrospec 3000 spectrophotometer (Pharmacia Biotech, Cambridge, England).

The content of endotoxin in AU701 and κ-carrageenan was 13.2 and 15.6 ng/mg, respectively, as measured using kinetic chromogenic Limulus amoebocyte lysate-test (Charles River Endosafe, INC, Reno, NV, USA). Lipopolysaccharide of *E. coli* 055:B5 (Charles River Endosafe, INC, Reno, NV, USA) was used for standard calibration curve.

### 3.2. Dynamic Oscillatory Measurements

For rheological measurements the pectin (P) solutions (1, 1.5 and 2%) and κ-carrageenan (Car 0.1, 0.2 and 0.5%) were well dispersed in deionized water. The dispersion was heated to 60 °C with magnetic stirring (200 rpm) in a water bath and held for 30 min, then cooled to 25 °C temperature in the tightly closed polyethylene bottle to ensure minimal evaporation during the process. The resulting solutions were transferred into dialysis tubes 4–5 cm long (dialysis tubing cellulose membrane. avg. flat width 33 mm (1.3 in.)). Then the dialysis tubes with polysaccharide solutions were placed in a solution containing 0.34 M calcium chloride and kept for 24 h to 25 °C. As a result, P-Car hydrogel samples were obtained in the form of cylinders 3–4 cm long followed by cutting off its cylindrical discs (diameter 2.8–3.0 cm, height 0.10–0.15 cm).

Oscillatory shear measurements of the storage and loss modulus (G’ and G”) and viscosity were performed using a controlled stress rheometer (Anton Paar, Physica MCR 302, Graz, Austria) equipped with parallel plate measurement systems (25 mm diameter, 1.0 mm gap). The sample loading area was preheated to 37 °C before hydrogel loading. After loading, the samples were equilibrated at 37 °C for 5 min before the measurement. The obtained mechanical spectra were characterized by the values of G’ and G” (Pa) as a function of frequency in the range of 0.05–20.00 Hz at 37 °C. The loss factor tan δ was calculated as the ratio of G” and G’. All experiments were performed at least in triplicate. 

The degree of frequency dependence for the G’ was determined by the power-law parameters [83,84], which is expressed as follows an equation: G’ = A*ω*^B^,(1)
where G’ is the storage modulus, *ω* is the oscillation frequency (Hz), and A is a constant.

### 3.3. Preparation of Gel Beads

The pectin-carrageenan gel beads were prepared using the ionotropic gelation method as described earlier [85]. Pectin and κ-carrageenan powders were weighted and mixed according to Table 5 and then dissolved in 10 mL of distilled water heated to 70 °C with stirring on a magnetic stirrer. The solutions were extruded using a nozzle with a 0.5-mm inner diameter into a 0.34 M calcium chloride solution (30 mL) with gentle agitation at room temperature. The distance from the nozzle to the calcium chloride solution was 5 cm. Thus, the gel beads were formed using an excess of calcium ions. The gel beads formed were allowed to stand in the solution for 30 min, followed by separation, and the beads were washed thrice with distilled water and dried at 37 °C for 18–20 h.

### 3.4. Characterization of Gel Beads

Images of dry gel beads (*n* = 40) were obtained using an optical microscope (Altami, Russia) equipped with a camera. The projected equivalent diameter of the beads was determined using an image analysis system (ImageJ 1.46r program, National Institutes of Health, Bethesda, MD, USA) with a calibration of 0.024 mm to one pixel.

A compression test of the gel beads was performed using TA-XT Plus Texture Analyser (Texture Technologies Corp., Stable Micro Systems, Godalming, UK). The wet gel beads were compressed at 25 °C with a 12 mm diameter (P/0.5R) cylinder probe with the test speed of 0.1 mm/s until deformation of 50%. The calculations of maximum peaks were performed for ten replicate samples using Texture Exponent 6.1.4.0 software (StableMicro Systems, Godalming, UK).

### 3.5. Swelling Characterization of Gel Beads

The dry gel beads (10 beads) were immersed in PBS or distilled water or cell culture medium (Hanks’ balanced salts solution supplemented with 25 mM HEPES (pH 7.4) and 10% foetal calf serum) for 2, 4, 8 and 24 h, respectively. The beads were shaken (100 rpm) in an orbital shaker incubator (Titramax 1000, Heidolph, Germany) at 37 °C. After a predetermined time interval, the projected equivalent diameter of beads was measured using an optical microscope (Altami, Russia) fitted with a camera and an image analysis system (Altami Studio, Altami, Russia). An image of a linear scale was used for calibration under the same optical conditions. One pixel corresponds to 0.00593 mm. The swelling degree (SD) was calculated using the following an equation:SD% = ((S_1_ − S_0_)/S_0_) × 100,(2)
where S_1_ is the surface area of the bead after a determined contact time with the liquid and S_0_ is the initial surface area.

### 3.6. BSA Adsorption of Gel Beads

The dried gels beads were used for adsorption studies [86]. BSA (100 μg/mL) solution was prepared in distilled water at different pH 3, 5, and 7 by adding 0.1 M HCl or 0.1 M NaOH to investigate the effect of pH of the initial BSA solution on the adsorption capacity of the beads. Then, 10 mg of dried gel beads were added into flasks containing 10 mL of BSA solutions with desired pH values. The flasks were placed on a thermostatic shaker (Heidolph Incubator 1000) at 25 °C for 24 h and the rate of the shaker was adjusted at 90 rpm. Upon incubation, samples were centrifuged and the supernatant containing protein was collected. The amount of protein adsorbed was quantified by Lowry’s method. Standard calibration curve of different concentrations of BSA (0–0.1 mg/L was plotted. Based on this curve, the amount of adsorbed protein (q) was calculated using an equation:q = ((C_0_ − C_1_)/m) × V,(3)
where C_0_ is the initial BSA concentration (mg/L), C_1_ is the remaining BSA concentration, V is the volume of protein solution used (L), and m (g) is the weight of gels beads. 

### 3.7. Complement Activation Evaluation

Venous blood was collected from healthy volunteers into vacuum tubes (Improvacuter, Guangzhou Improve Medical Instruments, Guangzhou, China) after obtaining written informed consent in accordance with relevant guidelines and regulations. The protocol was approved by the Ethical Committee of the Komi Science Centre of the Russian Academy of Sciences. Complement activation evaluation was performed by measuring the C3a levels as previously described [67]. To measure the release of C3a as an indicator for complement activation, 0.25 mL of human blood was added to the dry hydrogel samples at a concentration of 4 mg/mL. Zymosan A (Sigma-Aldrich, St. Louis, MO, USA) at a final concentration of 0.100 mg/mL was used as positive control and pyrogen-free 0.9% saline (NaCl, 0.05 mL) as a negative control. The solutions of 2% apple pectin and 0.5% κ-carrageenan (0.05 mL) were added to the blood samples as the reference compounds. The blood samples were incubated at 37 °C for 2 h. Following incubation, the supernatants of whole blood were collected after centrifugation 400 g for 20 min at 4 °C (Micro 220R Hettich Zentrifugen, Tuttlingen, Germany). Obtained supernatants were frozen and stored at −40 °C for later analysis. An enzyme-linked immunosorbent assay (ELISA) kit (Human C3a ELISA kit, Hycult Biotech, Uden, The Netherlands) was used according to manufacturer’s instructions to determine the concentration of released C3a fragments in plasma, which was diluted 1:300 with C3a-Sample Diluent.

### 3.8. Haemolysis Ratio Determination

The haemolysis ratio was measured after the incubation of whole blood with hydrogels. The dried gel beads (4 mg/mL) were transferred into sterile 2 mL microcentrifuge tubes (Eppendorf, Leipzig, Germany) and 0.3 mL of blood was added to each tube and incubated for 1 h at 37 °C without stirring. The samples were then centrifuged (400 g, 20 min), and 0.1 mL of the supernatant was collected from the tubes. The OD was measured at 540 nm using a Power wave 200 reader (BioTek Instruments, Santa Clara, CA, USA). The distilled water (0.1 mL) or pyrogen-free 0.9% saline (NaCl, 0.05 mL) were used as a positive or negative control, respectively. The haemolysis ratio was calculated in accordance with the following an equation:Haemolysis ratio (%) = ((A_s_ − A_n_)/(A_p_ − A_n_)) × 100,(4)
where A_s_ is the absorbance of the hydrogel sample, A_p_ and A_n_ are the absorbance the positive control and the negative control, respectively.

### 3.9. Hydrogel Adhesion to Biological Tissue

The force of hydrogel adhesion to the rat serosa was measured to evaluate the bioadhesive properties of the gels [24]. Adhesion strength was measured using a texture analyser TA-XT plus (Stable Micro Systems, Godalming, UK). A gel sample was prepared by immersing cotton buds in polysaccharide solution for 10 min with subsequent incubation for 30 min in 0.3 M CaCl_2_. The probe compressed the rat abdominal wall at 50 mN compression force for 20 s. The force of probe separation from the tissue after 20 s of pressing with a load of 0.05 N was recorded and calculated using Exponent Stable MicroSystems (Version V6.1.4.0) (Godalming, UK). A photograph illustrating the serosa-hydrogel adhesion test is shown in Figure 9.

### 3.10. Peritoneal Macrophages Adhesion and Activation

Macrophages were obtained using lavage of the abdominal cavity of male BalbC mice (25–30 g) with PBS (5 mL). The cells were centrifuged in saline for 10 min at 400× *g* and then resuspended in Hanks’ balanced solution containing 25 mM HEPES (pH 7.4) and 10% foetal calf serum. For cell response, 1 mL of 10^6^ peritoneal macrophages were incubated with 2 mg of gel beads at 37 °C for 4 h in microcentrifuge tubes (Eppendorf, Leipzig, Germany). The number of adherent cells was defined as an equation: ((C_0_ − C_4_)/C_0_) × 100,(5)
where C_0_ and C_4_ represent the initial cell concentration and after 4 h of co-incubation with the beads, respectively. After 4 h of incubation, the cells were precipitated by centrifugation (10 min, 400× *g*), and the supernatant was separated for subsequent cytokine evaluation (Figure 10).

Purified and biotin-conjugated anti-mouse TNF-α antibodies (eBioscience Inc., San Diego, CA, USA) were used for a sandwich ELISA measurement of the TNF-α level in the supernatants collected. The level of IL-10 was determined with a mouse IL-10 Immunoassay kit (R&D Systems, Minneapolis, MN, USA). Purified TNF-α and IL-10 (PeproTech Inc., Rocky Hill, NJ, USA) were used for the calibration curve. The optical density (OD) of the samples run in duplicate was measured with spectrophotometer Power Wave 200 (BioTek Instruments, Winooski, VT, USA). 

Peritoneal macrophages lysates were obtained by treatment of cell precipitates using the solution of 50 mM Tris-HCl, 150 mM NaCl, 50 mM NaF, 5 mM EDTA, 0.1% SDS, 0.1% Triton X-100, 1 mM PMSF, 1 mM Na_3_VO_4_ supplemented with protease inhibitor cocktail (Sigma-Aldrich, St. Louis, MO, USA). Western blotting analysis of NF-κB, TLR4, and ICAM-1 was performed as described earlier [87].

### 3.11. Statistical Analysis

One-way ANOVA with Tukey’s honest significance test was applied to determine statistically significant differences. Values of *p* ≤ 0.05 were considered statistically significant.

## 4. Conclusions

In this study, the biocompatibility of hydrogel made of apple pectin was improved by κ-carrageenan. Dynamic oscillatory measurements indicated that pectin and pectin- κ-carrageenan hydrogels exhibited a viscoelastic and viscous behaviour. The calcium pectin gel beads adsorbed BSA, induced C3a complement protein release, induced 1–2% haemolysis, and stimulated the production of TNF-α by peritoneal macrophages in vitro. The addition of the κ-carrageenan to the gel formulations led to a decrease in the gel strength. The P2-Car0.2, P2-Car0.5, and P1.5-Car0.5 gel beads induced lower non-specific protein adsorption than pure pectin gel beads. P1.5-Car0.5 gel beads were also found to reduce complement activation and decrease TNF-α production and NF-κB activation in mouse peritoneal macrophages compared to pure pectin gel beads. There were no differences in TLR4 and ICAM-1 levels in macrophages treated with P and P-Car gel beads. Thus, the blend of 1.5% apple pectin and 0.5% κ-carrageenan forms the hydrogel, which may have potential applications in tissue engineering as decreasing the initial stages of foreign body response due to an addition of κ-carrageenan to the apple pectin gel.

## Figures and Tables

**Figure 1 marinedrugs-20-00094-f001:**
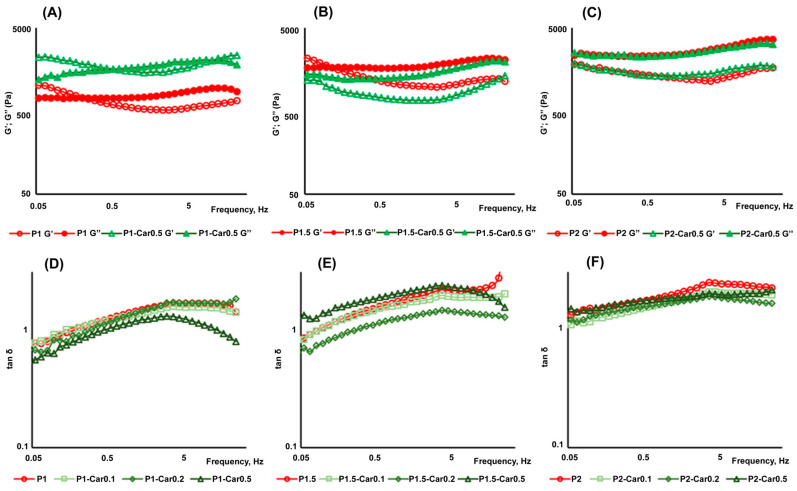
Rheological properties of P-Car hydrogels: storage modulus (G’) and loss modulus (G’’) test results (**A**–**C**); tan δ test results (**D**–**F**) represented as a function of frequency of P1 (circle symbols), P-Car0.1 (square symbols), P-Car0.2 (rhombus symbols) and P-Car0.5 (triangle symbols). Void symbols represent storage modulus G’, filled symbols represent viscous modulus G’’.

**Figure 2 marinedrugs-20-00094-f002:**
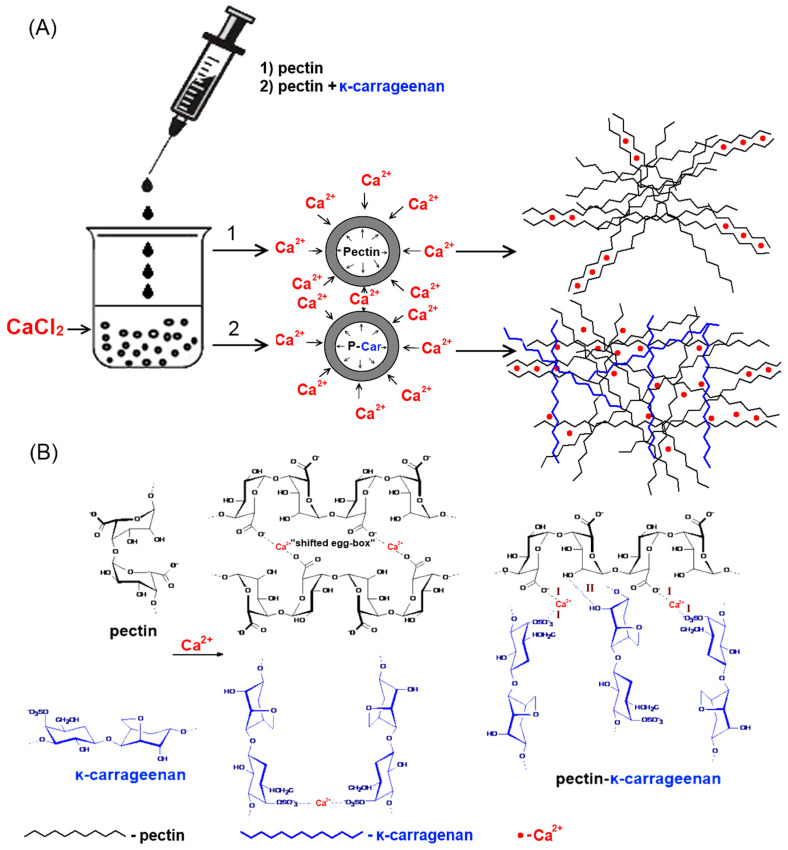
The method of preparation of P- and P-Car gel beads (**A**) and the mechanism presentation of induced pectin gelation with modulation by κ-carrageenan (**B**). (I)—ionic bonding; (II)—hydrogenic bonding.

**Figure 3 marinedrugs-20-00094-f003:**
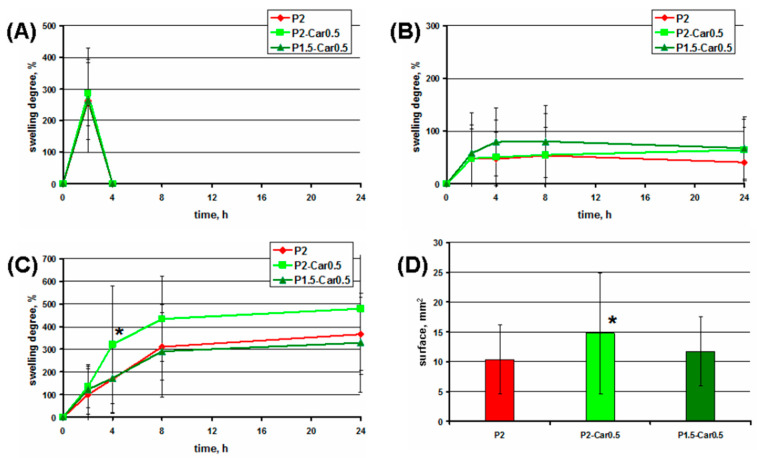
Swelling degree of gel bead incubating in PBS (**A**), distilled water (**B**), and cell culture medium (**C**). (**D**)—the average surface area of gel beads after 4 h incubation cell culture medium. The data are presented as the mean ± S.D. (*n* = 10). *—*p* < 0.05 vs. P2.

**Figure 4 marinedrugs-20-00094-f004:**
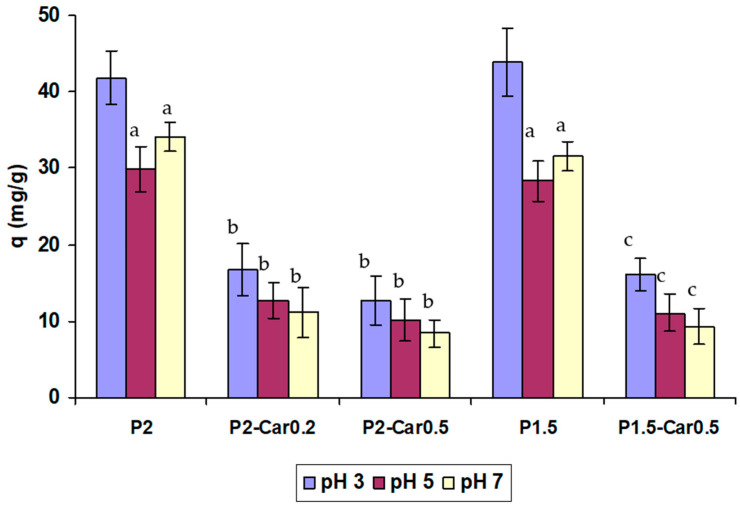
The BSA adsorption on the pectin and pectin-carrageenan gels at different pH. The data are presented as the mean ± S.D. (*n* = 6). ^a^ *p* < 0.05 vs. pH 3; ^b^ *p* < 0.05 vs. P2; ^c^ *p* < 0.05 vs. P1.5.

**Figure 5 marinedrugs-20-00094-f005:**
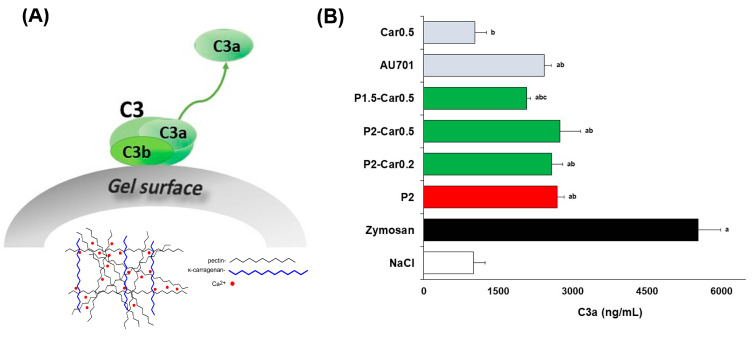
C3a production in the whole blood in vitro. Schematic presentation of C3a generation from C3 protein resulted from the contact with gel beads (**A**). Effect of pectin and pectin-carrageenan gel beads on the C3a production (**B**). The solutions of 2% apple pectin (AU701) and 0.5% κ-carrageenan (Car) were used as the reference compounds. Results are presented as the mean ± S.D. (*n* = 8). ^a^, ^b^, and ^c^—*p* < 0.05 vs. NaCl, Zymosan, and P2, respectively.

**Figure 6 marinedrugs-20-00094-f006:**
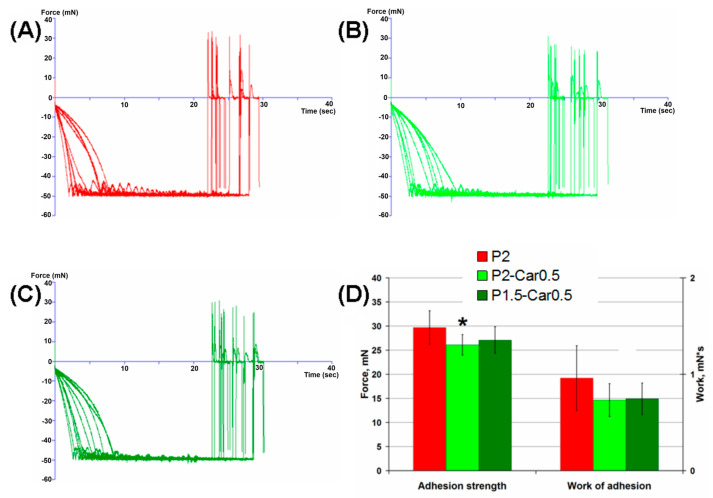
Representative adhesive curve of interaction of serosa with P2 (**A**), P2-Car0.5 (**B**), and P1.5-Car0.5 (**C**) hydrogels. (**D**)—force and work of hydrogel-serosa adhesion. Results are presented as the mean ± S.D. (*n* = 8). *—*p* < 0.05 vs. P2.

**Figure 9 marinedrugs-20-00094-f009:**
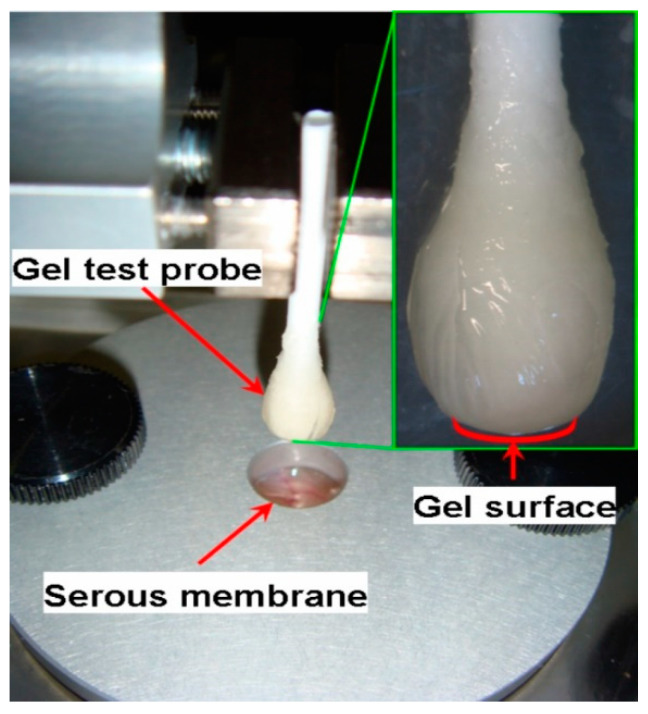
Schematic of serosa-hydrogel adhesion testing.

**Figure 10 marinedrugs-20-00094-f010:**
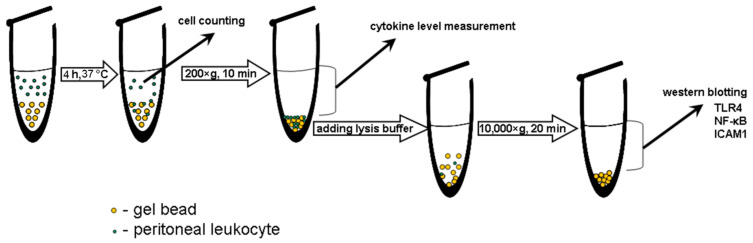
Schematic of testing peritoneal macrophages adhesion on the gel beads and activation.

**Table 1 marinedrugs-20-00094-t001:** Summary of power-law parameters for relationship between storage modulus and frequency (0.05 < *ω* < 20.00 Hz) of P-Car hydrogels.

Gels	Storage Modulus		
	A (Pa)	B (Slope)	R^2^
P1	629.7	−0.147	0.87
P1-Car0.1	891.5	−0.054	0.36
P1-Car0.2	1066.9	−0.027	0.05
P1-Car0.5	1732.2	−0.044	0.36
P1.5	1222.3	−0.151	0.82
P1.5-Car0.1	1493.0	−0.130	0.78
P1.5-Car0.2	2135.9	−0.100	0.75
P1.5-Car0.5	804.5	−0.042	0.15
P 2.0	1335.7	0.055	0.07
P2-Car0.1	1918.2	−0.089	0.56
P2-Car0.2	1805.8	−0.024	0.06
P2-Car0.5	1488.4	−0.013	0.02

**Table 2 marinedrugs-20-00094-t002:** Characterization of gel beads.

Gel Bead	Weight *, mg	Diameter *, mm	Gel strength **, N
P2	0.27 ± 0.05	1.08 ± 0.12	0.47 ± 0.06
P2-Car0.2	0.26 ± 0.04	1.13 ± 0.14	0.25 ± 0.04 ^a^
P2-Car0.5	0.37 ± 0.06 ^a^	1.14 ± 0.14	0.18 ± 0.03 ^a^
P1.5	0.19 ± 0.02 ^a^	1.44 ± 0.15 ^a^	0.28 ± 0.05 ^a^
P1.5-Car0.5	0.32 ± 0.03 ^b^	1.43 ± 0.13	0.11 ± 0.02 ^b^

*—dried beads; **—wet beads. The data are presented as the mean ± standard deviation (S.D.) ^a^ *p* < 0.05 vs. P2; ^b^ *p* < 0.05 vs. P1.5.

**Table 3 marinedrugs-20-00094-t003:** Effect of pectin and pectin-carrageenan gel beads on haemolysis of whole blood in vitro.

Sample	OD (540 nm)	Haemolysis Rate (%)
Distilled Water (Positive control)	2.634 ± 0.056 ^a^	100 ± 0
0.9% NaCl (Negative control)	0.074 ± 0.013	0 ± 0
P2	0.150 ± 0.037 ^a^	2.6 ± 0.96 ^a^
P2-Car0.2	0.123 ± 0.036 ^a^	2.1 ± 1.06 ^a^
P2-Car0.5	0.094 ± 0.040 ^ab^	0.8 ± 0.16 ^ab^
P1.5-Car0.5	0.100 ± 0.089 ^ab^	1.0 ± 0.35 ^ab^

OD—optical density. Values are presented as the mean ± S.D. (*n* = 8). ^a^ and ^b^—at *p* < 0.05 vs. negative control and P2, respectively.

**Table 4 marinedrugs-20-00094-t004:** Chemical characteristics of polysaccharides.

Sample	Sulphate (-SO_3_^−^) ^a^	UA ^a^	Gal ^a^	Xyl ^a^	Glc ^a^	Rha ^a^	Ara ^a^	OMe ^a^	DM ^a^	M_w_, kDa	M_w_/M_n_
AU701		86.5 ± 0.7	2.3 ± 0.1	2.8 ± 0.1	1.5 ± 0.1	1.3 ± 0.1	0.6 ± 0.4	6.2 ± 0.4	43	401	5.2
κ-Car	11.7 ± 0.7	-	29.3 ± 2.3	0.4 ± 0.2	9.4 ± 0.2	-	0.1 ± 0.0	-	-	1670	7.2

^a^ Data were calculated as weight %. UA—uronic acids, Gal—galactose, Xyl—xylose, Glc—glucose, Rha—rhamnose, Ara—arabinose, OMe—the amount of methyl groups, DM - degree of methyl esterification.

**Table 5 marinedrugs-20-00094-t005:** Preparation of gel beads *.

Sample Code	P2	P2-Car0.2	P2-Car0.5	P1.5	P1.5-Car0.5
Polysaccharide weight, mg					
Pectin	200	200	200	150	150
κ-carrageenan	-	20	50	-	50
Polysaccharide concentration, % *w*/*v*					
Pectin	2	2	2	1.5	1.5
κ-carrageenan	-	0.2	0.5	-	0.5

*—Polysaccharide powders of indicated weight were dissolved in 10 mL of distilled water.

## Data Availability

The data that support the findings of this study are available from the corresponding author upon reasonable request.

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
