# Peer review of "Characterization and Biocompatibility Properties In Vitro of Gel Beads Based on the Pectin and κ-Carrageenan"

_marinedrugs, 2022, doi:10.3390/md20020094_

Round 1

Reviewer 1 Report

Popov and colleagues studied the influence of kappa-carrageenan on the initial stages of the foreign body response against apple pectin gels. The study is interesting and well described, but the Authors should strenghten some results and revise some sentences.

  • Define acronyms (e.g., line 39: GalA). Use abbreviations (kappa-carrageenan vs κ-carrageenan).
  • Line 45: "The egg-box model describes the close packing of HG that occurs upon Ca2+-induced gelling". Several studies do not fully agree with this sentence (e.g., "Significant difference was indeed found between alginate and pectin in terms of Ca-binding and gelation", DOI: 10.1016/j.foodres.2018.08.20). Revise this sentence and Fig.2
  • Line 49: "However, pectin hydrogel may present challenges in tissue engineering due to foreign body response (FBR) which results in poor integration implant with native tissue". Add references to support this sentence.
  • Revise Paragraph 2.1. A) lines 86-88: "Gels are usually classified as strong or weak [25]. Strong gels have the characteristics of “true” gels when they exhibit the behavior of viscoelastic solids under small deformation, whereas they rupture rather than flow above a critical deformation value [26]". Why this definition is relevant? B) Line 93: "the values of G’ at the beginning being a little higher than G’’ then below, which is typical for viscous solutions". Check. C) Lines 95-96: "The spectra demonstrated G′ ≥ G″ or G′ ≤ G″ for all hydrogels when different kappa-carrageenan concentrations were used for the gel’s preparation", but in the following line the Authors described the effect of pectin, without mentioning kappa-carrageenan. Revise or remove. Lines 98-99 repeat previous sentences. D) Why did the Authors apply the analysis summarized in Table 1? Why is it relevant in the context of the manuscript? E) Does the results in Fig.1 and Table 1 depend on pectin/carrageenan concentration only or also on pH?
  • Table 2: Diameter*, mg. Check.
  • Lines 171-172: "First, the calcium ions contained in the cell culture medium can increase cross-linking and the strength of the gel". Could the Authors comment on this point with reference to Secchi et al., 2014 (DOI: 10.1088/0953-8984/26/46/464106)?  
  • Lines 271: "Apple pectin used in the present study to obtain gel beads consists mainly of the HG region". Rephrase.
  • The Authors should state which is the best combination of pectin and kappa-carrageenan to minimize the initial stages of the foreign body response.
  • For pectin, the most common extraction source are citrus fruits, but the Authors exploited pectin from apples. Could they extend their results to the combination of pectin from citrus fruits and kappa-carrageenan? Are the conclusions identical?
  • Paragraph Statistical analysis: Why did not the Authors perform group analysis (ANOVA test or equivalent for not normally distributed data) before comparing two means? 
  • English language is fine and minor changes are required (e.g., Line 72: "induce FBR did not earlier investigated"; line 418 "to be equal to 13.2 and 15.6 ng/mg of AU701 and kappa-carrageenan"; line 423 "the pectin’s (P) solutions".

Reviewer 2 Report

In my opinion, the manuscript is suitable for publication in Marine Drugs after the Authors have addressed some comments and questions. In summary, this is a good paper, but I have mixed feelings regarding the resulting materials and materials terminology. Are the Authors sure that gel beads based on the apple pectin and kappa-carrageenan are composite gel beads? The definitions of polymer blend and composites are quite contradictory, although they are often used together. However, according to available literature, they are not the same. Please explain why the Authors call the studied beads composites? There are abbreviations without explanation in a few places (e.g., ECM, GalA) - please correct and complete them.  In section "3. Materials and Methods" - "3.3. Preparation of composite gel beads" - in addition to the procedure for obtaining the beads, methods of testing them are described herein. Please separate the process for obtaining the beads from their testing methods.  

Round 2

Reviewer 1 Report

I would like to thank Popov and colleagues to provide a revised version of their manuscript. Their rebuttal replied to most of my comments. However, I have still some minor comments:

1) Lines 99-102: A confirmation of the viscoelastic state of gels cross-linked by ion Ca2+ was obtained when tan δ = 1 (G’≥ G’’) (Figure 1 D, E, and F). All the hydrogels prepared had the tan δ values at low frequencies close to 1, further increasing the frequency tan δ ≥ 1.  Please, check. tan δ = 1 means G’= G’’.

2) Line 103: The data obtained indicate that P and P-Car hydrogels possess a “viscoelastic gel” structure. Why viscoelastic gel and not viscelastic?
